# Vasoplegic Syndrome after Cardiac Surgery for Infective Endocarditis

**DOI:** 10.3390/jcm11195523

**Published:** 2022-09-21

**Authors:** Pascal Lim, Margaux Le Maistre, Lucas Benoudiba Campanini, Quentin De Roux, Nicolas Mongardon, Valentin Landon, Hassina Bouguerra, David Aouate, Paul-Louis Woerther, Fihman Vincent, Adrien Galy, Vania Tacher, Sébastien Galien, Pierre-Vladimir Ennezat, Antonio Fiore, Thierry Folliguet, Raphaelle Huguet, Armand Mekontso-Dessap, Bernard Iung, Raphael Lepeule

**Affiliations:** 1Service de Cardiologie, DMU Médecine, Assistance Publique-Hôpitaux de Paris (AP-HP), Hôpitaux Universitaires Henri Mondor, Faculté de Santé, Univ. Paris Est Créteil, F-94010 Créteil, France; 2Service d’anesthésie-Réanimation Chirurgicale, Assistance Publique-Hôpitaux de Paris (AP-HP), Hôpitaux Universitaires Henri Mondor, Faculté de Santé, Univ. Paris Est Créteil, F-94010 Créteil, France; 3Laboratoire de Bactériologie et Virologie, Assistance Publique-Hôpitaux de Paris (AP-HP), Hôpitaux Universitaires Henri Mondor, Faculté de Santé, Univ. Paris Est Créteil, F-94010 Créteil, France; 4Unité Transversale de Traitement des Infections, DMU PDTI, Assistance Publique-Hôpitaux de Paris (AP-HP), Hôpitaux Universitaires Henri Mondor, Faculté de Santé, Univ. Paris Est Créteil, F-94010 Créteil, France; 5Service de Radiologie, Assistance Publique-Hôpitaux de Paris (AP-HP), Hôpitaux Universitaires Henri Mondor, Faculté de Santé, Univ. Paris Est Créteil, F-94010 Créteil, France; 6Service de Maladies Infectieuses et Immunologie, Assistance Publique-Hôpitaux de Paris (AP-HP), Hôpitaux Universitaires Henri Mondor, Faculté de Santé, Univ. Paris Est Créteil, F-94010 Créteil, France; 7Service de Chirurgie Cardiaque, DMU CARE, Assistance Publique-Hôpitaux de Paris (AP-HP), Hôpitaux Universitaires Henri Mondor, Faculté de Santé, Univ. Paris Est Créteil, F-94010 Créteil, France; 8Service de Médecine Intensive Réanimation, DMU Médecine, Assistance Publique-Hôpitaux de Paris (AP-HP), Hôpitaux Universitaires Henri Mondor, Faculté de Santé, Univ. Paris Est Créteil, F-94010 Créteil, France; 9Service de Cardiologie, Assistance Publique-Hôpitaux de Paris (AP-HP), Hôpitaux Universitaires Bichat et Université Paris Cité, Assistance Publique-Hôpitaux de Paris (AP-HP), F-75018 Paris, France

**Keywords:** endocarditis, vasoplegic syndrome, shock, outcome

## Abstract

Purpose: Post-operative vasoplegic syndrome is a dreaded complication in infective endocarditis (IE). Methods and Results: This retrospective study included 166 consecutive patients referred to cardiac surgery for non-shocked IE. Post-operative vasoplegic syndrome was defined as a persistent hypotension (mean blood pressure < 65 mmHg) refractory to fluid loading and cardiac output restoration. Cardiac surgery was performed 7 (5–12) days after the beginning of antibiotic treatment, 4 (1–9) days after negative blood culture and in 72.3% patients with adapted anti-biotherapy. Timing of cardiac surgery was based on ESC guidelines and operating room availability. Most patients required valve replacement (80%) and cardiopulmonary bypass (CPB) duration was 106 (95–184) min. Multivalvular surgery was performed in 43 patients, 32 had tricuspid valve surgery. Post-operative vasoplegic syndrome was reported in 53/166 patients (31.9%, 95% confidence interval of 24.8–39.0%) of the whole population; only 15.1% (*n* = 8) of vasoplegic patients had a post-operative documented infection (6 positive blood cultures) and no difference was reported between vasoplegic and non-vasoplegic patients for valve culture and the timing of cardiac surgery. Of the 23 (13.8%) in hospital-deaths, 87.0% (*n* = 20) occurred in the vasoplegic group and the main causes of death were multiorgan failure (*n* = 17) and neurological complications (*n* = 3). Variables independently associated with vasoplegic syndrome were CPB duration (1.82 (1.16–2.88) per tertile) and NTproBNP level (2.11 (1.35–3.30) per tertile). Conclusions: Post-operative vasoplegic syndrome is frequent and is the main cause of death after IE cardiac surgery. Our data suggested that the mechanism of vasoplegic syndrome was more related to inflammatory cardiovascular injury rather than the consequence of ongoing bacteremia.

## 1. Introduction

Endocarditis is defined as an inflammation of the endocardial surface of the heart. In most cases, the inflammation is caused by a bacterial infection and the most frequent causative pathogens are streptococci, staphylococci or enterococci. Infective endocarditis (IE) is a serious disease with a stable incidence of 30–100 episodes per million patients-years [1]. The mortality rate remains high despite diagnosis and therapeutic improvements. In addition to antibiotic therapy, surgical treatment is required in approximately half of the patients with IE. Reasons to consider early surgery in the active phase (i.e., while the patient is still receiving antibiotic treatment) are to control infection, to avoid progressive heart failure (HF) and irreversible structural damage, and to prevent systemic embolism [2]. Although this aggressive therapeutic strategy may represent the last chance to save life and eradicate the infection in patients with uncontrolled bacteriemia or heart failure, the benefit of early surgery on mortality is still a matter of debate in hemodynamically stable patients. Randomized studies are scarce and limited by sample size population but results from most of studies are in favor of early surgical management in complicated IE [3]. The European Society of Cardiology (ESC) Guidelines provide clear recommendations for the surgical indications during the early phase of the disease [4]. These indications must be balanced with patient comorbidities and operative risk because postoperative mortality remains particularly high, between 6% to 25%. The most dreaded post-operative complication remains refractory shock, with vasoplegia being one of the main mechanisms. Vasoplegic syndrome is supposed to be related to an excessive inflammatory response and endothelial nitric oxide release initiated by the sepsis and heart failure and further exacerbated by the cardiopulmonary bypass [5,6]. Post-operative vasoplegic syndrome had been investigated in routine cardiac surgery, but no specific data has been reported in IE.

## 2. Methods

### 2.1. Population Study

The Henri Mondor university hospital has developed a dedicated multidisciplinary endocarditis team, with expertise over a large part of Greater Paris area. The endocarditis team included at least cardiologists, cardiac surgeons, infectiologists, bacteriologists, radiologists, and intensive care physicians. All patients with high suspicion of endocarditis are hospitalized in cardiac intensive care to benefit from a fast multidisciplinary diagnostic and therapy strategy. The endocarditis team reviews every week the medical strategy of all suspected or confirmed endocarditis diagnosis. In addition, all confirmed endocarditis cases are followed at least one year after the diagnosis by the endocarditis team. We retrospectively studied all consecutive patients hospitalized for definite or possible acute left or right-sided IE referred to cardiac surgery (*n* = 243) between 2016 and 2021. We excluded 50 cardiac device lead extractions without cardiopulmonary bypass required during the procedure and 27 patients in cardiogenic or septic shock before cardiac surgery. The final IE diagnosis status was determined according to the ESC 2015 modified diagnostic criteria and [4], the perioperative findings and the 6-month follow-up. All patients provided a written informed consent to participate to the study.

Transthoracic (TTE) and trans-esophageal (TEE) echocardiography were systematically performed in all patients at admission. Echocardiographic data were systematically digitally stored for off-line analysis. Left ventricular ejection fraction was quantified using Simpson biplane method. Right ventricular function was graded by using tricuspid annular plane systolic excursion (TAPSE) measurement. The severity of valve stenosis or regurgitation was assessed according to the current guidelines and, globally EROA (Effective Regurgitation Orifice Area) by the PISA approach was used to grade the severity of valvular regurgitation [7]. The larger size of vegetation was considered for the analysis. Abscess was suspected in the presence of hypoechoic spherical or thickness structure and was confirmed by cardiac CT imaging or surgical finding.

Computed tomography imaging (CT) protocol for endocarditis included a total body CT-cardiac coupled with an ECG gated cardiac CT [8]. CT was systematically performed in all patients hemodynamically stable unless renal or allergic contra-indication. Brain magnetic resonance imaging and cerebral artery angiogram were performed in addition to brain-CT in case of cerebral bleeding on CT-brain or in patients with a contra-indication to contrast injection. Patients with cardiac implantable electronic devices or heart valve prostheses underwent PET-CT when TEE and cardiac-CT were non-conclusive.

Bacteriological examinations included repeated blood cultures at admission [9]. Blood samples were systematically collected for Coxiella burnetti and Bartonella spp. serologies. Prosthetic valve, cardiac devices, vegetations and valvular tissue were sent to bacteriological laboratory for culture. PCR with 16s RNA gene sequencing was performed in patients with negative blood or cardiac tissue cultures.

Endocarditis treatment was standardized by the endocarditis team that included cardiologists, infectiologists, intensive care physicians, anesthesiologists and cardiac surgeons. The first line antimicrobial therapy followed the current ESC guidelines and was secondary adapted to the bacteriological finding [4]. Cardiac surgery indication was based on the ESC guidelines after assessing the operative risk using EuroSCORE-II. The decision was taken after consensus within the endocarditis team. The timing to send patient to cardiac surgery was based on the ESC guidelines but also on the avaibility of operative room. Preventive functional valve intervention on non-endocarditis valve (mitral or tricuspid) was discussed before the intervention.

Post-operative vasoplegic syndrome was defined by defined by the following criteria: (1) the need of norepinephrine administration (>0.3 mg/h for more than 12 h) despite fluid expansion to maintain a mean arterial pressure ≥ 65 mmHg, (2) a cardiac index ≥ 2.2 L/min/m^2^, or a preserved left ventricular ejection fraction, or a central venous blood saturation (ScVo2 ≥ 60%), and (3) the absence of documented infection [6]. Septic shock was defined by the following criteria: (1) the need of vasopressor to maintain a mean arterial pressure ≥65 mmHg in the absence of hypovolemia, (2) signs of organ dysfunction defined by serum lactate level > 2 mM/L, and (3) a documented infection with a positive blood or valve cultures. Transient post-operative vasoplegia that required temporary (<12 h) and/or low dose of norepinephrine (<0.3 mg/h) was not classified as a post-operative vasoplegic syndrome. Hypoxic hepatitis was defined as a sudden and significant increase of aspartate aminotransferase (AST > 5 times the upper limit of normal) in response to cardiac, circulatory or respiratory failure and after exclusion of other causes of liver injury [10]. Low cardiac output was treated by dobutamine support with the rate adjusted to maintain Scvo2 ≥ 60% [11]. Cardiac assistance (ECMO) was indicated in patient’s refractory to maximum dose of catecholamine.

### 2.2. Statistical Analysis

Continuous variables with a normal distribution were expressed as mean ± SD, while non-normally distributed variables were expressed as median and quartiles (25th and 75th). Normality distribution was graphically assessed for continuous variables. Nominal variables were expressed using percentages. Comparison between groups was performed by using Student test or variance analysis for continuous variables and by X^2^ for percentages. Variables independently associated with the onset of postoperative vasoplegic syndrome were identified by stepwise multivariable logistic regression that included variables with *p* < 0.1 from univariate analysis. Stepwise multivariable linear regression was used to identify variables independently associated with cardiopulmonary bypass duration. Two-tailed *p* values < 0.05 were considered to indicate statistical significance. Statistical analyses were performed using StatView version 5.0 and SPSS for Windows (IBM, Armonk, NY, USA).

## 3. Results

### 3.1. Patients

Of 166 patients included (64 ± 14 years, 77% male, Table 1 and Table 2), the majority (89%) had a definite IE diagnosis based on modified Duke criteria before cardiac surgery and all were classified as definite after surgical findings. IE lesions occured on the native valve in 69.3% of patients. Blood cultures were positive in 159 (95.8%) of patients, Streptococcus spp. was found in 65 (39.9%) patients, Staphylococcus spp. in 54 (32.5%) patients [36 of Staphylococcus aureus] and, Enterococcus spp. in 23 (13.8%) patients. Vegetation and abscess were identified in 137 (82.5%) and 47 (28.3%) of patients (*n* = 24 (14.4%) by CT), respectively. The mean vegetation size was 15 ± 6 mm, and severe mitral or aortic valvular regurgitation was reported in half of patients (48.8%, *n* = 81). At admission, one-third (32.5%, *n* = 54) of patients had heart failure symptoms. Stroke was reported in 74 (44.6%) patients: 69 had ischemic (27 with bleeding complications) and 5 had pure hemorrhagic cerebral bleeding. Finally, the mean EuroScore-II before surgery was 9.8 ± 3.6 (Table 1 and Table 2).

### 3.2. Cardiac Surgery

Cardiac surgery was performed 7 days (5–12) after the beginning of antibiotic therapy with appropriate antibiotic therapy in 120/166 (72.3%, Table 3). Blood culture were still positive the day or the day before surgery in 9/166 patients (5.4%). Valve replacement was required in most of patients (80.1%, *n* = 133). Multivalvular surgery was performed in 43 (25.9%) patients, and 29 (17.4%) patients required both mitral and aortic valve interventions. Tricuspid valve surgery was performed in 32 patients, 7 were isolated tricuspid surgery and 25 combined with a left side surgery (11 combined with both mitral and aortic surgery, 14 combined with either mitral or aortic surgery). Most of combined tricuspid surgery (*n* = 23/25, 92%) were performed to correct functional tricuspid regurgitation. Median CPB duration was 133 (95–184) minutes. Variables independently associated with CPB duration were prosthesis related IE, the presence of abscess and multi-valvular surgery. Culture of vegetation, abscess, cardiac device or valvular lesions available in 112 patients were positive in 44% (*n* = 51).

### 3.3. Vasoplegic Syndrome

Vasoplegic syndrome was reported in 53 (31.9%) patients (31.9%, 95% confidence interval of 24.8–39.0%) and all occurred within the first day following cardiac surgery. Bacterial infection was documented in 8 (15.1%) patients, mainly in blood (*n* = 6), sputum (*n* = 6) and mediastinal sample (*n* = 3) culture. Noradrenalin (4.9 mg/h (2.0–15.0) vs. 0.4 mg/h (0.1–0.8)) and dobutamine (5.0 µg/kg/min (0.0–7.3) vs. 0 µg/kg/min (0.0–5.0)) was more required in the vasoplegic group (Table 2). Hemusiccinate was used in 6 vasoplegic patients and none in non-vasoplegic group. Scvo2 under catecholamine averaged 71 ± 9% in vasoplegic vs. 67 ± 12% in non-vasoplegic group (*p* = 0.06). Right atrial pressure was reported in 164 patients and was >10 mmHg in 46.2% of vasoplegic group vs. 18.8 in non-vasoplegic group (*p* = 0.0002). Veino-arterial ECMO support (*n* = 12) was mainly implanted in the vasoplegic group (83%, 10/12). Finally, renal replacement therapy (26.4% vs. 13.3%, *p* = 0.04), mechanical ventilation duration, hypoxic hepatitis, and major bleeding were more reported in the vasoplegic group (Table 3). In-hospital death occurred in 23 patients (13.8%) during hospitalization and most of them were reported in the vasoplegic group (87%, *n* = 20/23). The first cause of death in the vasoplegic group was multi-organ failure (87%, *n* = 20/30) and neurological complications (*n* = 3). The cause of death in patients without vasoplegic syndrome (*n* = 3) were neurological complications.

Univariate variables (Table 1, Table 2 and Table 3) associated with post-operative vasoplegic syndrome were older age and EuroSCORE-II. Vasoplegic group had more frequent abscess (Table 2), lower LVEF (55 ± 8% vs. 58 ± 7%, *p* = 0.05), more severe right ventricular dysfunction (17.0% vs. 6.2%, *p* = 0.03), higher NT-pro-BNP (3788 pg/mL vs. 1998 pg/mL, *p* = 0.0005) and troponin level. Albumin and creatinine clearance were lower in vasoplegic group. Cardiac surgical characteristics included a greater rate of multivalvular surgery (37.7% vs. 20.3%) and tricuspid valve surgery (32.1% vs. 13.3%, *p* = 0.006), and a more prolonged cardiopulmonary bypass (169 (122–237) min vs. 121 (89–164), *p* = 0.001) and cardioplegia duration in the vasoplegic group. No difference was observed for the rate of positive valve culture or the timing of cardiac surgery relative to the beginning of antibiotherapy or the end of positive blood culture.

Finally, variables independently associated with post-operative vasoplegic syndrome included Nt-Pro-BNP (OR = 2.11 (1.35–3.30) for each tertile, *p* = 0.001, Table 4) and cardiopulmonary bypass duration (0R = 1.82 (1.16–2.88) for tertile, *p* = 0.009). These independent variables remains unchanged even after excluding the 8 patients with a documented infection. Figure 1 shows the risk of refractory vasoplegia according to cardiopulmonary bypass duration in patients hemodynamically stable at admission.

## 4. Discussion

The study shows that vasoplegic syndrome occurs in more than on third of patients after cardiac surgery for acute IE. Importantly, post-operative deaths (*n* = 23) were almost all reported in the vasoplegic group (87%, 20/23). Preoperative variables independently associated with the risk of vasoplegia included preoperative NT-proBNP and cardiopulmonary bypass duration.

The vasoplegic syndrome was first described by Gomes et al. [12] after cardiac surgery. The cardiac vasoplegic syndrome is a form of vasodilatory shock that occurs after surgery with cardiopulmonary bypass. The main explanation for this hemodynamic scenario is a systemic inflammatory response secondary to the release and the activation of proinflammatory cytokines causing a generalized vasodilatation. This syndrome, which is characterized by a severe and persistent arterial hypotension, decreased systemic vascular resistance and normal or increased cardiac output, requires catecholamine agents and is associated with high mortality [13]. The incidence of vasoplegic syndrome for routine cardiac surgery was estimated at 5% in a recent meta-analysis including >30,000 patients [14]. Risk factors associated with vasoplegic syndrome included renal failure before cardiac surgery, previous cardiac surgery, preoperative use of antihypertensive medications, cardiopulmonary bypass and aortic cross clamp duration, combined surgery and amount of red blood cell transfusion. Our study is the first addressing this dreaded complication in patients referred to cardiac surgery for IE. We found that the incidence of post-operative vasoplegic syndrome (31.9%) is particularly high. The identification and prevention of this syndrome is probably one of the key strategy for improving IE outcome because most of death occurred in this subgroup of patients.

Interestingly, patients developing vasoplegic syndrome rarely had post-operative positive blood culture (11%, *n* = 6/53) and no greater marker of uncontrolled sepsis (valve culture, timing of surgery regarding negative blood culture) compared to patients who did not developed vasoplegic syndrome. Independent variable associated with vasoplegic syndrome were only elevated Nt-proBNP level and CPB duration. This suggests that active infection is probably not the direct mechanism of vasoplegic syndrome but rather a trigger of vasoplegic susceptibility. Increase in NT-pro-BNP level during IE is multifactorial related to the severity of valvular lesion and to the sepsis-induced cardiac dysfunction. Sepsis related to IE is accompanied by local and strong systemic inflammatory response with elevated circulating IL-6, IL-2R and IL-1β concentrations [15]. These cytokines together with a variety of bacteria-derived products (e.g., formylated chemotactic peptides, lipopolysaccharides) may enhance myocardial ROS generation that contributes to cardiac dysfunction independently to IE-induced valve damage [16,17,18]. Heart failure treatment optimization is often attempted but the therapeutic margin is often limited and prompt surgery is usually the unique solution in patients with severe valvular lesions.

Cardiopulmonary bypass duration was consistently reported as a strong risk factor of vasoplegic syndrome [12,13,14]. Reducing cardiopulmonary bypass is complex because most of factors cannot be controlled excepted for combined surgery. In the setting of endocarditis, straightforward operative strategy targeting only severe valvular lesions should probably be preferred over an exhaustive strategy. Indeed, preventive functional surgery on the mitral or tricuspid and aortic root surgery should be postponed and balanced with the risk of vasoplegia. Similarly, the strategy between valve replacement or repair should be balanced with surgical expertise to avoid prolonged surgical duration.

Finally, the therapeutic margin for optimizing the heart failure treatment and cardiopulmonary bypass duration is probably weak in the setting of emergency surgery. The innovative therapy would be counteracting or preventing systemic inflammatory that appears to be the main mechanism of vasoplegic syndrome [19,20]. Inhibiting inflammatory process during elective or urgent cardiac surgery with high dose of dexamethasone (1 mg/kg) has been evaluated in a large randomized double-blind placebo-controlled study to prevent vasoplegic syndrome [20]. The primary outcome (death, myocardial infarction, stoke, renal or respiratory failure at 30 days) was similar to placebo but dexamethasone was associated with reductions in post-operative infection, duration of post-operative mechanical ventilation and length of intensive care unit and hospital stay. However, there is no specific study in endocarditis patients, probably because the use of immunosuppressive therapy is generally not recommended during active bacterial infection even though several authors have reported safety and successful outcome in patients with IE under corticosteroids. Methyl-prednisolone (0.5 g daily for 3 days followed by 30, 20 and 10 mg for the consecutive 3 days) was correlated to normalization of serum inflammatory markers and restoration of renal insufficiency in a patient with IE dependent glomerulonephritis [21]. Successful administration of prednisone (60–80 mg daily) in 3 patients with immune mediated renal insufficiency and in 2 patients with Austrian syndrome was also reported [22]. The benefit of corticosteroids is probably related to cytokine production modulation by the inhibition of transcription factors as nuclear factor kB (NFKB) and activated protein 1 of inflammatory prostaglandins and of lymphocytes apoptosis [23]. In a methicillin resistant Staphylococcus aureus (MRSA) animal experimental IE model, a combination of vancomycin plus corticosteroids was associated with less severe histopathological valve lesions compared to treatment with vancomycin alone [24]. In a methicillin-sensitive Staphylococcus aureus (MSSA) IE animal model, the addition of dexamethasone to antimicrobial therapy significantly reduced blood TNF alpha levels compared to the control.

## 5. Limitations

Independent variables associated with vasoplegic syndrome were derived from our single center study population and should be validated in an independent cohort. However, we believe that our results are robust because most of variables associated to the risk of vasoplegic syndromes are consistent with those reported in routine cardiac surgery. This robustness may be related to the large sample size study and the standardization of the diagnosis and therapy strategy that may contribute to reduce bias. However, this standardization may also limit the use of potential benefit drug or device such as extracorporeal blood purification device [25] or methylene blue, which was not currently used because of the lack of robust clinical data. Recent propensity score-matched cohort study in patients who developed vasoplegic syndrome during cardiac surgery showed that methylene blue reduced the cumulative amount of vasopressor but without modifying patient’s outcome [26]. The prophylactic use of methylene blue in elective IE surgery was assessed in a small sample (*n* = 42) randomized double-blind placebo-controlled study [27]. The administration of the methylene blue before CPB did not change the post-operative needs of vasoconstrictor drugs.

## 6. Perspectives

Regarding the high incidence of vasoplegic syndrome and its associated mortality in endocarditis surgery, there is an urgent need of investigating the potential benefit of steroids therapy combined or not with an extracorporeal blood purification device as a preventive therapy of post-operative vasoplegia.

## 7. Conclusions

This is the first study reporting incidence and risk factors associated with post-operative vasoplegic syndrome in the endocarditis population. Our study showed that more than one third of patients experienced vasoplegia and importantly almost all post-operative death occurred in this group of patients (*n* = 20/23). Variables independently associated with the risk of vasoplegia were not the timing of surgery or positive valve culture but cardiopulmonary bypass duration and preoperative NT-proBNP value. These data highlighted the need to develop preventive strategy to reduce the risk of vasoplegia.

## Figures and Tables

**Figure 1 jcm-11-05523-f001:**
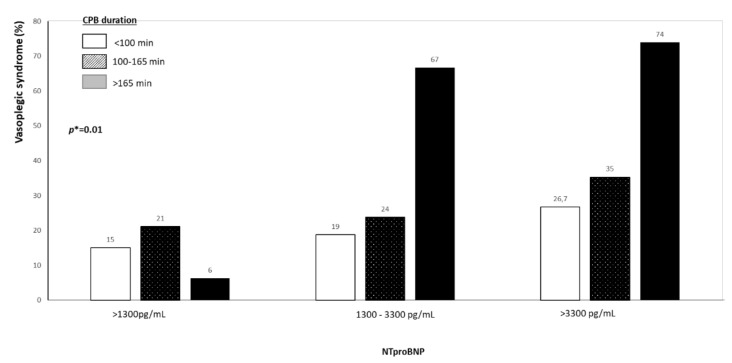
Incidence of vasoplegic syndrome according to NtproBNP value and cardio-pulmonary bypass duration, *p** indicates the interaction term between Nt-proBNP and CPB duration.

**Table 1 jcm-11-05523-t001:** Clinical characteristics of the population.

	All (*n* = 166)	No Vasoplegia (*n* = 113)	Vasoplegic Syndrome (*n* = 53)	*p*
Age, years	64 ± 14	62 ± 15	69 ± 14	0.005
Gender, M	128 (77.1)	88 (77.8)	40 (75.5)	0.73
History of stroke, *n* (%)	5 (3)	2 (1.8)	3 (5.7)	0.17
BMI, kg/m^2^	26 ± 6	26 ± 6	25 ± 9	0.87
Hypertension, *n* (%)	90 (54.2)	58 (51.3)	32 (60.4)	0.27
Atrial fibrillation, *n* (%)	34 (20.5)	21 (18.6)	13 (24.5)	0.38
Diabete mellitus, *n* (%)	43 (25.9)	28 (24.8)	15 (28.3)	0.63
History of heart failure, *n* (%)	12 (7.2)	6 (5.3)	6 (11.3)	0.16
Coronary artery disease, *n* (%)	29 (17.5)	19 (16.8)	10 (18.9)	0.75
History of endocarditis, *n* (%)	16 (9.6)	12 (10.6)	4 (7.5)	0.53
History of cardiac surgery, *n* (%)	65 (39.2)	32 (28.3)	33 (41.5)	0.09
Cirrhosis, *n* (%)	5 (3.0)	4 (3.5)	1 (1.9)	0.56
COPD, *n* (%)	6 (3.6)	2 (1.8)	4 (7.5)	0.06
Dialysis, *n* (%)	3 (1.8)	2 (1.8)	0 (0.0)	0.20
EuroSCORE II (%)	9.8 ± 3.6	8.7 ± 3.2	10.6 ± 3.2	0.0006

**Table 2 jcm-11-05523-t002:** Clinical characteristics of the population.

	All (*n* = 166)	No Vasoplegia (*n* = 113)	Vasoplegic Syndrome (*n* = 53)	*p*
			Doc Sepsis (*n* = 53)	
Fever, *n* (%)	126 (76.0)	86 (76.1)	40 (75.5)	0.93
Congestive heart failure, *n* (%)	53 (31.9)	33 (29.2)	20 (37.7)	0.27
Arterial embolism, *n* (%)	76 (45.6)	52 (46.0)	24 (45.3)	0.85
Ischemic stroke, *n* (%)	69 (41.6)	49 (43.4)	20 (37.7)	0.49
Hemorragic stroke, *n* (%)	32 (19.2)	24 (21.2)	8 (15.1)	0.35
Positive hemoculture, *n* (%)	159 (95.8)	107 (94.7)	52 (98.1)	0.30
Staphy aureus, *n* (%)	36 (21.7)	24 (21.2)	12 (22.6)	0.84
Staphy epidermidis, *n* (%)	18 (10.8)	9 (8.0)	9 (7.5)	0.93
Streptococcus spp, *n* (%)	65 (39.9)	45 (39.8)	20 (37.7)	0.80
Enterococcus spp., *n* (%)	23 (13.8)	14 (12.4)	9 (17.0)	0.71
Bartonella spp., *n* (%)	6 (3.6)	3 (2.7)	3 (5.7)	0.33
C-Reactive Protein, mg/mL	74 (38–117)	74 (33–115)	74 (45–130)	0.11
Uremia, mmol/L	7.9 ± 5.5	7.3 ± 5.5	9.0 ± 5.5	0.07
Creat Clear, mL/min/1.73 m^2^	76 ± 31	79 ± 31	68 ± 8	0.02
NT-proBNP, pg/mL	2240 (1021–4379)	1998 (845–3319)	3788 (1868–8928)	0.0005
Troponin I, ng/L	37 (15–103)	28 (14–64)	68 (24–208)	0.001
Total bilirubin, μmol/L	10 (7–17)	10 (7–15)	10 (7–15)	0.85
Arterial lactate, mmol/L	1.2 (1.2–1.2)	1.2 (1.2–1.2)	1.2 (1.2–1.2)	0.85
Albumin, g/L	28 ± 6	29 ± 6	27 ± 5	0.005
Echocardiograpy Data				
Vegetation, *n* (%)	137 (82.5)	95 (84.1)	42 (79.2)	0.44
Vegetation size, mm (*n* = 163)	15 ± 6	14 ± 5	16 ± 6	0.14
Abcess, *n* (%)	47 (28.3)	25 (22.1)	22 (41.5)	0.01
Severe MR, *n* (%)	39 (23.6)	26 (23.2)	13 (24.5)	0.85
Severe AR, *n* (%)	49 (29.5)	33 (29.2)	18 (30.2)	0.45
Severe TR, *n* (%)	14 (8.4)	8 (7.1)	6 (11.3)	0.36
LVEF, %	57 ± 8	58 ± 7	55 ± 8	0.05
TAPSE < 15 mm, *n* (%)	16 (9.6)	7 (6.2)	9 (17.0)	0.03
CT and PET data				
Abcess by CT, *n* (%)	24 (14.4)	11 (9.7)	13 (24.5)	0.01
PET fixation (*n* = 36), *n* (%)	27 (69.4)	16 (66.7)	11 (91.7)	0.11
IE status before surgery				
Definite, *n* (%)	147 (89.0)	98 (86.7)	50 (94.3)	0.14
Prothesis IE, *n* (%)	51 (30.7)	30 (26.5)	21 (39.6)	0.09
Lead IE, *n* (%)	12 (2.0)	3 (5.3)	6 (11.3)	0.16
Native valve IE, *n* (%)	115 (69.3)	82 (72.6)	33 (62.3)	0.18
Mitrale valve IE, *n* (%)	78 (47.0)	54 (47.8)	24 (45.3)	0.76
Aortic valve IE, *n* (%)	102 (61.4)	65 (57.5)	37 (69.8)	0.13
Tricuspide IE, *n* (%)	9 (5.4)	6 (5.3)	3 (5.7)	0.93
Multivalvular, *n* (%)	26 (15.7)	17 (14.2)	10 (18.9)	0.44

**Table 3 jcm-11-05523-t003:** Operative and post-operative patient’s characteristics.

	All (*n* = 166)	No Vasoplegia (*n* = 113)	Vasoplegic Syndrome (*n* = 53)	*p*
Adapted anti-biotherapy	120 (72.3)	83 (73.5)	37 (69.8)	0.63
Antibiotic starting to surgery, days	7 (5–12)	7 (5–12)	8 (5–14)	0.45
Negative BC to surgery, days	4 (1–9)	4 (2–9)	4 (1–9)	0.93
Cardioplegia duration, min	133 (95–184)	121 (89–164)	169 (122–237)	0.001
CPB duration, min	106 (74–143)	97 (72–130)	126 (86–164)	0.01
Valve replacement, *n* (%)	133 (80.1)	86 (76.1)	47 (88.7)	0.06
Aortic valve surgery, *n* (%)	107 (64.3)	68 (60.0)	39 (73.6)	0.10
Mitral valve surgery, *n* (%)	79 (47.5)	53 (46.8)	26 (49.1)	0.79
Tricuspid valve surgery, *n* (%)	32 (19.2)	15 (13.3)	17 (32.1)	0.006
Multivalve surgery, *n* (%)	43 (25.9)	23 (20.3)	20 (37.7)	0.02
Lead extraction, *n* (%)	18 (9.3)	9 (8.3)	9 (9.4)	0.80
Positive valve culture (*n* = 112), *n* (%)	51 (45.5)	32 (43.2)	19 (50)	0.50
Blood red cell unit, *n*	2.0 (0.0–3.0)	2.0 (0.0–2.0)	2.0 (2.0–5.0)	0.0003
Platelet unit, *n*	0.0 (0.0–1.0)	0.0 (0.0–0.0)	0.0 (0.0–1.0)	0.04
Fresh Frozen Plasma, *n*	0.0 (0.0–3.0)	0.0 (0.0–2.0)	0.0 (0.0–4.0)	0.22
Intensive care data				
SOFA score	8.2 ± 2.6	7.9 ± 2.6	8.9 ± 2.7	0.03
Renal replacement therapy, *n* (%)	29 (17.5)	15 (13.3)	14 (26.4)	0.04
Mechanical ventilation, days	0 (0–2)	0 (0,1)	2 (1–8)	0.03
Hypoxic hepatitis (*n* = 164), *n* (%)	21 (12.8)	5 (4.4)	16 (31.4)	<0.001
Hemisuccinate, *n* (%)	6 (3.6)	0 (0)	6 (11.3)	0.002
Noradrenaline, mg/h	0.7 (0.2–3.0)	0.4 (0.1–0.8)	4.9 (2.0–15.0)	<0.0001
Dobutamine, µg/kg/min	0.0 (0.0–5.0)	0.0 (0.0–0.0)	5.0 (0.0–7.3)	0.003
ECMO support, *n* (%)	12 (7.4)	2 (1.8)	10 (19.6)	0.0002
Scvo2, % (*n* = 138)	68 ± 11	67 ± 12	71 ± 9	0.06
RAP > 10 mmHg, *n* = 164	45 (27.5)	21 (18.8)	24 (46.2)	0.0002
Arterial lactate, mmol/L	1.4 (1.1–2.0)	1.3 (1.1–1.6)	1.9 (1.2–3.8)	<0.0001
C-Reactive Protein, mg/mL	78 (51–126)	82 (58–127)	65 (47–115)	0.11
Re-intervention	26 (15.8)	7 (6.2)	19 (36.5)	<0.0001
Tamponnade or bleeding, *n* (%)	31 (19)	9 (8.0)	32 (44.0)	<0.0001
Valve dysfunction, *n* (%)	6 (3.6)	4 (3.5)	2 (4.0)	0.88
Mediastinitis, *n* (%)	6 (3.6)	2 (1.8)	4 (7.8)	0.06
In hospital death, *n* (%)	23 (13.8)	3 (2.6)	20 (37.7)	<0.0001

**Table 4 jcm-11-05523-t004:** Variables independently associated with the risk of vasoplegia (multivariate analysis).

Variables	OR (95% CI)	*p*
Preoperative NT-proBNP (per tertile)	2.11 (1.35–3.30)	0.001
CPB duration (per tertile)	1.82 (1.16–2.88)	0.009
Tricuspid surgery	2.17 (0.91–5.19)	0.08

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
