# Peer review of "Vasoplegic Syndrome after Cardiac Surgery for Infective Endocarditis"

_jcm, 2022, doi:10.3390/jcm11195523_

Round 1

Reviewer 1 Report

This is an interesting retrospective study of incidence and predictors of vasoplegic syndrome after heart surgery in patients with infective endocarditis. Although findings are not unexpected, information in this topic is scarce and the authors did a great job in implementing the major clinical variables that may afftect the outcome.

I have only a few minor comments:

Was the reported EF preop or postop? Post op variables such as EF, valvlular leak/stenosis and presence of unaddressed valvular lesions should be implemented in analysis

Author Response

The authors would thank the Reviewer for the relevancy of his reviewing.

Question 1: Was the reported EF preop or postop? Post op variables such as EF, valvlular leak/stenosis and presence of unaddressed valvular lesions should be implemented in analysis.

We agree with the Reviewer and reported in the Table 3 the value of post-operative mixed oxygen central venous saturation that better reflects the balance between supply and need than left ventricular ejection fraction (LVEF). In addition, post operative echocardiography was only performed  in unstable patients and LVEF was usually semi-quantatively assessed because ICU physicians are not trained to Simpson biplane method. As required, we also reported the rate of post-operative valve dysfunction (3.6%), which was similar between the two groups.  

Reviewer 2 Report

Thank you for the opportunity to review this manuscript.

In this manuscript, the authors investigate the characteristics and outcomes of patients undergoing surgery for infective endocarditis with a focus on patients that developed vasoplegia following cardiac surgery. The manuscript is well written, thorough, and documents the characteristics and outcomes of 166 patients who underwent cardiac surgery for infective endocarditis at a single center. The authors identified trends with increased age, EuroSCORE II, cardiopulmonary bypass time, troponin, albumin, and BNP levels being more common in patients that developed postoperative vasoplegia, as well as vasoplegia being predictive of mortality in this cohort.

Major Comments

1.     Overall, the Discussion section is strong and well written. The authors discuss previous and ongoing research as well as some of the clinical implications of this research. I would only make one suggestion regarding the Discussion, and that is to discuss future directions for research on this topic. It would tie the Discussion together and would be of interest to readers if the authors were to briefly propose unanswered questions regarding this topic and how future studies could effectively address them. If the authors are able to make this addition, I feel this Discussion will be very strong overall.

 Minor Comments

1.     In the abstract, the authors state that cardiac surgery was performed 7 days after beginning antibiotic treatment and 4 days after negative blood culture. Was this on average or is this the standard protocol for every patient in this cohort? This difference should be clarified in the abstract.

2.     In the abstract, the authors state “only 15.1% (n=8) of vasoplegic patients had a documented infection (6 positive blood cultures)”. While it can be assumed that since this manuscript is discussing patients with infective endocarditis that this statement is referring to a documented postoperative infection, this is not specified. It should be clarified that the authors are referring to cases of postoperative infection, at the time of the development of vasoplegia.

3.     In the Methods, the authors note that their center has developed a dedicated multidisciplinary endocarditis team, although no further details are provided. This will likely be of interest to readers and I would encourage the authors to describe this team in greater detail including the specialists and allied health care staff involved, as well as briefly how the team functions at their center.

4.     In Table 1, the column with EuroSCORE has brackets with (%), although it does not seem that a % value is provided for this column. The authors should remove the (%) from this column or add in a corresponding value.

5.     In Tables 2 and 3, there are values provided in rounded or square brackets. The round In brackets are defined as % in the left column of the row they are associated with, while the square brackets are not defined. It should be clear somewhere on the Table whether the square brackets are referring to the absolute range, interquartile range, 95% confidence interval, or some other range.

6.     The tables and figures provided are appropriate and complement the information provided in this study. 

Author Response

Major Comments

  1. Overall, the Discussion section is strong and well written. The authors discuss previous and ongoing research as well as some of the clinical implications of this research. I would only make one suggestion regarding the Discussion, and that is to discuss future directions for research on this topic. It would tie the Discussion together and would be of interest to readers if the authors were to briefly propose unanswered questions regarding this topic and how future studies could effectively address them. If the authors are able to make this addition, I feel this Discussion will be very strong overall.
Response: we thank the Reviewer for his remark and this has been modified in the revised manuscript.  

 Minor Comments

  1. In the abstract, the authors state that cardiac surgery was performed 7 days after beginning antibiotic treatment and 4 days after negative blood culture. Was this on average or is this the standard protocol for every patient in this cohort? This difference should be clarified in the abstract
This is the median time interval and this has been clarified in the abstract and method section of the revised manuscript.
  1. In the abstract, the authors state “only 15.1% (n=8) of vasoplegic patients had a documented infection (6 positive blood cultures)”. While it can be assumed that since this manuscript is discussing patients with infective endocarditis that this statement is referring to a documented postoperative infection, this is not specified. It should be clarified that the authors are referring to cases of postoperative infection, at the time of the development of vasoplegia.

We agree with the Reviewer and this has been clarified in the abstract.

  1. In the Methods, the authors note that their center has developed a dedicated multidisciplinary endocarditis team, although no further details are provided. This will likely be of interest to readers and I would encourage the authors to describe this team in greater detail including the specialists and allied health care staff involved, as well as briefly how the team functions at their center.

We totally agree and this has been precised in the revised manuscript.

  1. In Table 1, the column with EuroSCORE has brackets with (%), although it does not seem that a % value is provided for this column. The authors should remove the (%) from this column or add in a corresponding value.

This is the predict mortality and we corrected.

  1. In Tables 2 and 3, there are values provided in rounded or square brackets. The round In brackets are defined as % in the left column of the row they are associated with, while the square brackets are not defined. It should be clear somewhere on the Table whether the square brackets are referring to the absolute range, interquartile range, 95% confidence interval, or some other range
The authors agree with the Reviewer and this has been clarified.

1- The tables and figures provided are appropriate and complement the information provided in this study.

The authors thank the Reviewer for his appreciation.